# Germline Whole-Gene Deletion of FH Diagnosed from Tumor Profiling

**DOI:** 10.3390/ijms22157962

**Published:** 2021-07-26

**Authors:** Arisa Ueki, Kokichi Sugano, Kumiko Misu, Eriko Aimono, Kohei Nakamura, Shigeki Tanishima, Nobuyuki Tanaka, Shuji Mikami, Akira Hirasawa, Miho Ando, Teruhiko Yoshida, Mototsugu Oya, Hiroshi Nishihara, Kenjiro Kosaki

**Affiliations:** 1Clinical Genetic Oncology, Cancer Institute Hospital, Japanese Foundation for Cancer Research, 3-8-31 Ariake, Koto-ku, Tokyo 135-8550, Japan; 2Center for Medical Genetics, Keio University Hospital, 35 Shinanomachi, Shinjuku-ku, Tokyo 160-8582, Japan; ksugano4608@titan.ocn.ne.jp (K.S.); kumiko.misu@adst.keio.ac.jp (K.M.); kkosaki@keio.jp (K.K.); 3Genomics Unit, Keio Cancer Center, Keio University Hospital, 35 Shinanomachi, Shinjuku-ku, Tokyo 160-8582, Japan; eriko0123@keio.jp (E.A.); knakamura320@keio.jp (K.N.); hnishihara1971@keio.jp (H.N.); 4Genome Center, Department of Cancer Prevention & Genetic Counseling, Tochigi Cancer Center, 4-9-13 Younan, Utsunomiya-shi, Tochigi 320-0834, Japan; miho.ando1103@gmail.com; 5Biomedical informatics Development Department, Mitsubishi Space Software Co., Ltd., Amagasaki, Hyogo 661-0001, Japan; Tanishima.Shigeki@mss.co.jp; 6Department of Urology, Keio University School of Medicine, 35 Shinanomachi, Shinjuku-ku, Tokyo 160-8582, Japan; urotanaka@keio.jp (N.T.); moto-oya@z3.keio.jp (M.O.); 7Department of Diagnostic Pathology, Keio University Hospital, 35 Shinanomachi, Shinjuku-ku, Tokyo 160-8582, Japan; mikamishuji@gmail.com; 8Department of Clinical Genomic Medicine, Graduate School of Medicine, Dentistry and Pharmaceutical Sciences, Okayama University, 2-5-1 Shikata, Kita-ku, Okayama 700-8558, Japan; hir-aki45@okayama-u.ac.jp; 9Division of Genetics, National Cancer Center Research Institute, 5-1-1 Tsukiji, Chuo-ku, Tokyo 104-0045, Japan; tyoshida@ncc.go.jp

**Keywords:** hereditary leiomyomatosis and renal cell carcinoma, fumarate hydratase, tumor profiling, germline genetic testing, exome sequencing

## Abstract

Hereditary leiomyomatosis and renal cell carcinoma (HL (RCC)) entails cutaneous and uterine leiomyomatosis with aggressive type 2 papillary RCC-like histology. HLRCC is caused by pathogenic variants in the FH gene, which encodes fumarate hydratase (FH). Here, we describe an episode of young-onset RCC caused by a genomic FH deletion that was diagnosed via clinical sequencing. A 35-year-old woman was diagnosed with RCC and multiple metastases: histopathological analyses supported a diagnosis of FH-deficient RCC. Although the patient had neither skin tumors nor a family history of HLRCC, an aggressive clinical course at her age and pathological diagnosis of FH-deficient RCC suggested a germline FH variant. After counseling, the patient provided written informed consent for germline genetic testing. She was simultaneously subjected to paired tumor profiling tests targeting the exome to identify a therapeutic target. Although conventional germline sequencing did not detect FH variants, exome sequencing revealed a heterozygous germline FH deletion. As such, paired tumor profiling, not conventional sequencing, was required to identify this genetic deletion. RCC caused by a germline FH deletion has hitherto not been described in Japan, and the FH deletion detected in this patient was presumed to be of maternal European origin. Although the genotype-phenotype correlation in HLRCC-related tumors is unclear, the patient’s family was advised to undergo genetic counseling to consider additional RCC screening.

## 1. Introduction

Hereditary leiomyomatosis and renal cell carcinoma (HL(RCC)) is an inherited syndrome that causes multiple cutaneous leiomyomas, severely symptomatic uterine leiomyomas, and papillary type 2 RCC-like histology owing to anomalies in the FH gene that encodes fumarate hydratase (FH). The severity of HLRCC varies significantly within and between families. Although the penetrance of HLRCC-related RCC is not considered high, the prognoses of patients with this progressive condition are reportedly poor. For example, 9 of 13 patients in a particular study reportedly died of renal cancer metastasis within five years of their diagnosis [1].

Smit et al. reported that the penetrance of cutaneous leiomyoma in their cohort was almost 100% after age 40 and that 11 of 21 women who were FH pathogenic variant carriers (52%) were treated for uterine myomas by the age of 35 years [2]. They also reported that type 2 RCC and Wilms tumors developed in 2 of 35 FH pathogenic variant carriers. These data suggested that, while the penetrance of RCC is not high, the aggressiveness of the malignancy is. As such, they proposed clinical diagnostic criteria for HLRCC; their major indicator was histopathologically confirmed multiple cutaneous piloleiomyomas, while the minor criteria were: (1) surgical treatment for severely symptomatic uterine leiomyomas before the age of 40 years, (2) type 2 papillary RCC before age 40, and (3) a first-degree family member who meets one of the abovementioned criteria [2]. Given that multiple uterine myomas are very often observed in the general population, it can be difficult to narrow down the diagnosis based on multiple uterine myomas alone; however, type 2-like FH deficient RCC is considered to be typical for HLRCC despite its low penetrance.

The FH gene controls energy metabolism, and FH is involved in the tricarboxylic acid (TCA) cycle in mitochondria (Figure 1). The enzymatic function of FH is required for the conversion of fumarate. When FH malfunctions, the enzymatic activity of FH decreases, causing fumarate to accumulate. This, in turn, leads to succinate accumulation, stabilizing the hypoxia-inducible factor 1-alpha subunit, and causing the upregulation of vascular endothelial growth factor (VEGF) and glucose transporter 1 (GLUT1), thereby contributing to the carcinogenesis of high-grade RCC [3].

In FH-deficient RCC, oxidative phosphorylation is impaired, and aerobic glycolysis is upregulated (a phenomenon referred to as the Warburg effect). Energy production in cancer cells thereby becomes almost fully dependent on glycolysis, the increase of which amplifies adenosine triphosphate production, which in turn promotes cell proliferation. Through this mechanism, VEGF (promoting angiogenesis) and GLUT1 (increasing glucose transport) are upregulated. Both these events underlie the molecular mechanism of high-grade RCC [4]. FH-deficient RCC is often detected by positron emission tomography-computed tomography. The RCC components of HLRCCs carrying a pathogenic variant of the FH gene are thus considered to be highly malignant and have poor prognoses.

FH is not included in the list of genomic “secondary findings” devised by the American College of Medical Genetics and Genomics (ACMG) in 2017 [5]. The original germline findings comprised a list of 56 genes referred to as “incidental findings” by the ACMG in 2013 [6]. The list was updated to version 2.0 in 2017 to include 59 genes (referred to as ACMG59 or ACMG ver2.0), and recently in 2021, updated to version3.0 to include 73 genes (referred to as ACMG73 or ACMG ver3.0) [7]. The ACMG73 genes are considered clinically manageable, and 29 of them are listed for their association with hereditary tumors, including *BRCA1/2* and Lynch syndrome-causative genes. The European Society of Medical Oncology (ESMO) Precision Medicine Working Group recommended germline-focused analysis of tumor-only sequences in 2019, and FH was included among the 27 targeted genes [8]. At Keio University Hospital, we have already encountered patients with FH-related RCC, and we currently include FH as a gene to be screened for during tumor profiling tests.

Molecular tumor profiling is an epochal method for determining therapeutic targets. Precision medicine has recently been advocated, and personalized treatment strategies based on tumor profiling have attracted attention. As a concern for tumor profiling, the possibility of hereditary cancer-related genes can be confirmed by comparison with paired normal tissues as controls. The ACMG (2020) issued a statement on presumed germline pathogenic variants that can be revealed by tumor profiling tests [9]. They stated three tumor profiling strategies as follows: (1) tumor-only testing, (2) tumor-normal paired testing with germline variant subtraction, and (3) tumor-normal paired testing with a full analysis of the germline data from a subset of genes associated with cancer predisposition [9]. We have previously developed a clinical sequencing system named “PleSSision Exome test” to perform a comprehensive tumor-normal paired profiling [10]. PleSSision Exome test analyzes both tumor and germline DNA and reports germline-derived pathogenic variants as germline findings.

By disclosing these germline findings, useful information can be provided to patients and their families, appropriate surveillance methods for early detection and early treatment for cancers can be suggested, indications for risk-reducing surgery can be discussed, and appropriate treatment for individual patients can be selected [11].

Here, we describe a patient with early-onset RCC caused by a genomic FH deletion that was diagnosed via tumor profiling analysis.

## 2. Case Presentation

### 2.1. Patient’s Information and Clinical Course

A 35-year-old woman with no previous history of disease was diagnosed with RCC and multiple metastases while undergoing examination for abdominal pain. Renal biopsy was performed, and histopathological analyses supported the diagnosis of FH-deficient RCC (Figure 2).

The patient was diagnosed with stage-4 RCC as well as multiple metastases in the lung, liver, supraclavicular lymph nodes, para-aortic lymph nodes, adrenal glands, and retroperitoneum. Combination therapy with ipilimumab and nivolumab (a standard treatment for RCC) was started immediately after diagnosis. After three doses of the combination therapy, the symptoms worsened, marked accumulation of ascites was noted, and cancer-related pain was exacerbated. Hence, the patient was subjected to a PleSSision Exome test to determine the therapeutic target.

Although the patient had neither skin tumors nor a family history of HLRCC (Figure 3), an aggressive clinical course during her 30s suggested the possibility of a germline FH variant. Loss of FH protein expression in the renal biopsy sample also suggested HLRCC.

After genetic counseling, written informed consent for germline genetic testing was obtained from the patient. However, at the time of tumor profiling, her RCC was already advanced and rapidly progressing, and she eventually died five months after diagnosis and four months after the start of treatment. Her death occurred before the test results were available; the large FH deletions found in the germline were disclosed to her family.

### 2.2. Genetic Testing Results

Although conventional germline sequencing did not detect pathogenic FH variants, exome sequencing for tumor profiling revealed a heterozygous germline genetic deletion of FH. As PleSSision Exome test is a tumor-normal paired profiling test, genomic deoxyribonucleic acid (DNA) was extracted and purified from tumor tissues as well as peripheral blood mononuclear cells (PBMCs) obtained from the patient. Exome sequencing can detect not only single nucleotide variants (SNVs) but also copy number variants (CNVs), enabling a wide range of diagnoses. A large homozygous deletion of approximately 3 Mb near FH was confirmed in the tumor sample near the end of chromosome 1q as observed using the copy number alteration plot of the tumor-normal paired test (Figure 4). As HLRCC was originally suspected based on the phenotype, the germline CNV confirmed a large deletion of the FH gene.

When the copy number was tested in another panel for confirmation, results showed that the FH gene was not expressed at all in the tumor; there was a homozygous FH deletion at copy number = 0 in the tumor, and a heterozygous FH deletion, observed as copy number = 1 in the germline (Appendix A).

When re-examining the deleted region in the tumor-normal paired exome analysis results in detail, the loss in CNV was detected to be limited to the FH gene’s vicinity. However, the variant allele frequency (VAF) plot showed that the loss of heterozygosity (LOH) or acquired uniparental disomy (UPD) might have occurred in the entire 1q chromosome in the tumor. In other words, the VAF was split more widely in chromosome 1q, suggesting that a LOH or acquired UPD had occurred in the tumor. Among the regions where the VAF plot split, three suspected deletions were found in the CNV (at q23.3, q31.3, and q43, including the FH gene) (Appendix A).

These data led to the postulate that there were originally three LOH regions in the germline FH-deficient allele. In the process of carcinogenesis, a large deletion occurred in the healthy allele, and by copying the FH-deficient allele, 3 LOH regions appeared in both alleles in the cancerous tissue cells, resulting in acquired somatic UPD (Figure 5).

Next-generation germline sequencing alone did not reveal any pathogenic FH variants. However, upon tumor profiling using the multiplex ligation-dependent probe amplification (MLPA) method, extensive germline FH deletion was confirmed, leading to the diagnosis of HLRCC (Figure 6a). Moreover, the region containing the nearby *EXO1* gene was also found to be deleted; this was consistent with the tumor profiling data (Figure 6b). These findings confirmed that the germline FH gene had a large deletion and that the patient was afflicted with HLRCC.

### 2.3. Materials and Methods

Immunohistochemical analysis was performed on formalin-fixed paraffin-embedded tissues using the FH mouse monoclonal antibody (J-13: sc-100743, 1:200 dilution; Santa Cruz Biotechnology, Santa Cruz, CA, USA). In the PleSSision Exome test, genomic DNA was extracted and purified from tumor tissues and PBMCs obtained from the patient. Gene expression profiling was performed using the MiSeq sequencing platform (Illumina). Genome annotation and curation for analyzing the sequencing data were performed using the original bioinformatics pipeline GenomeJack (Mitsubishi Space Software, Tokyo, Japan) as previously described [10]. GRCh37/hg19 was used for data analysis as the human reference genome assembly in the PreSSision Exome test. Germline panel testing using the OncoGuide NCC Oncopanel System FC v. 1.0 (Agilent, Tokyo, Japan) was performed as previously described [12], and GRCh38/hg38 was used for data analysis as the human reference genome assembly. MLPA using a kit purchased from MRC-Holland (Amsterdam, the Netherlands) was performed as previously described, according to the manufacturer’s recommendations [13], and GRCh37/hg19 was used for data analysis as the human reference genome assembly.

## 3. Discussion

Although large FH deletions have previously been reported, they are relatively rare in HLRCC. In a review of patients registered in the Leiden Open Variation Database (LOVD) published in 2008 [14], 93 of 107 variants were considered pathogenic and were noted for their genetic location. These FH variants, which we also considered pathological in this report, are dispersed throughout the FH locus rather than being concentrated at the N-terminus. Phenotypic sequelae such as RCC onset are diverse, and no unequivocal genotype-phenotype correlations associated with certain types of variants or their locations in the FH gene have emerged to date [14].

HLRCC disease onset and fumarase deficiency are more likely to be associated owing to the *difference* in the gene expression level rather than the *position* of the FH variant. Most variants are missense, nonsense, and frameshift mutations, but four percent are large deletions that can only be detected by MLPA [14]. Such large deletions include whole exon and whole-gene deletions.

We re-investigated the updated LOVD [15] and examined 396 FH gene variants, 150 of which were found to be pathogenic or likely pathogenic. These variants were not limited to nonsense and frameshift mutations, as missense mutations accounted for a majority (55%) of the variants (Figure 7). Large deletions, as in our patient, accounted for five percent, which was similar to the rate reported in 2008.

In our patient, a large deletion could not be detected (HLRCC may have been overlooked) when using conventional sequencing methods. Using the MLPA method first described in France in 2008, 19 patients who had previously been clinically diagnosed with HLRCC but had no pathological variants detected using direct sequencing were re-examined [16]. An exon 1 deletion was only detected in a single patient. Widespread deletions containing the FH gene have been reported for some time but are relatively rare. Another French group, the French National Cancer Institute “Inherited Predispositions to Kidney Cancer” network, also reported patients with complete FH deletions, albeit less frequently: they reported 4 complete deletions of FH, 1 exon deletion, 1 exon duplication, and 81 different FH germline point mutations among 144 HLRCC families [17]. It is necessary to devise a way to avoid overlooking the large FH deletions, although the frequency is low.

Several previously reported Japanese families with HLRCC exhibited different pathogenic variants, including missense variants and a splice site variant that had not been reported in Caucasian families [18,19,20,21,22,23]. To the best of our knowledge, ours is the first report of a whole-gene FH deletion in Japan. The patient’s mother (from whom the FH gene deletion was posited to originate) is from the United Kingdom, which is consistent with several existing reports of FH gene deletions in Europeans. Therefore, the whole-gene FH deletion detected in this patient was presumed to be of European origin. Our new annotated variants have been deposited to the MGeND database (https://mgend.med.kyoto-u.ac.jp/mgend.med.kyoto-u.ac.jp, accessed on 26 July 2021.), which is a ClinVar-type, open-access database for use by the broad scientific community.

Although we could not administer a genetic test to the patient’s younger brother because he lived abroad, we provided the family a letter with genetic test results and surveillance recommendations. Smit et al. reported that 100% of FH pathogenic variant carriers over the age of 40 years had cutaneous leiomyomas [2]. In a previous study by Alam et al., cutaneous leiomyomas were observed in 89% of FH pathogenic variant carriers, including all (100%) of the men [24]. Such leiomyomas had a particularly high penetrance; however, different phenotypes were observed within the family investigated in Alam et al.’s study. While cutaneous leiomyomas can be painful and may require treatment, surveillance for RCC remains the higher priority.

FH-related RCC is known to have a poor prognosis, as was the case for our patient. Muller et al. found that the lifetime risk of developing RCC in French FH pathogenic variant carriers is 19% [25]. In 2011, Smit et al. recommended semi-annual renal ultrasonography and annual magnetic resonance imaging (MRI) starting at the age of 20 years [2]. To date, 12 patients younger than 20 years with FH-related RCC have been reported, and their tumors were aggressive in nature [26]. Therefore, it has been proposed that screening should start at a younger age. In 2013, the Second Symposium on Hereditary Leiomyomatosis and Renal Cell Cancer recommended genetic testing for FH in children within affected families starting from the ages of 8–10 years onward, with those positive for the pathogenic variant to commence annual screening with renal MRI [27]. Owing to the autosomal dominance of HLRCC, it was important to alert our patient’s younger brother (age 32 years) because he has a 50% chance of carrying the same variant as his sister.

Several treatments, including mTOR inhibitors, are available for the treatment of FH-VAF-deficient RCC, but evidence of their effectiveness is insufficient [27]. Considering the energy metabolism pathway in FH-deficient carcinoma shown in Figure 1, we expect a therapeutic strategy that will lead to the prevention and complete response of FH-deficient carcinoma. Further investigations ought to clarify the carcinogenic mechanism of FH-deficient RCC and produce more effective FH-targeting treatments.

## 4. Conclusions

The germline generic FH deletion in our patient was detected not via conventional sequencing but paired tumor profiling. RCC caused by a germline FH gene deletion has hitherto not been described in Japan, and FH deletion detected in this patient was presumed to be of maternal European origin. Although the genotype-phenotype correlation in patients with HLRCC-related tumors is unclear, the patient’s family was advised to undergo genetic counseling regarding HLRCC for further screening of renal malignancies. If there are no known patients with HLRCC in a family, the diagnosis of type 2 papillary RCC and/or FH-deficient RCC before the age of 40 should raise suspicions of this familial pathogenic variant. Finally, recent significant advances in screening technologies have improved the detection of large deletions upon genetic profiling of tumors; however, it is necessary to know the limits of each testing method when making a genetic diagnosis.

## Figures and Tables

**Figure 1 ijms-22-07962-f001:**
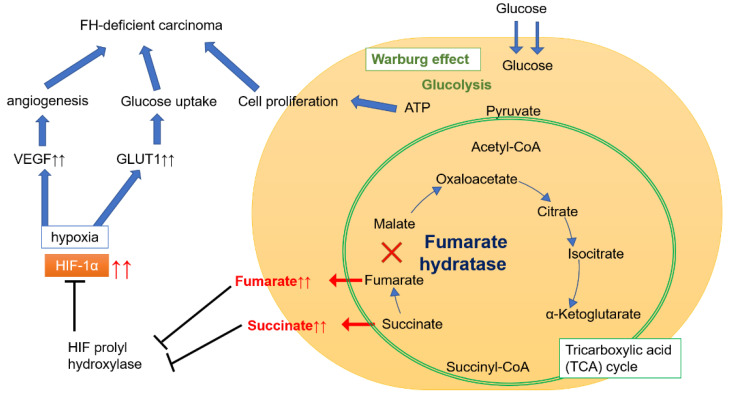
Energy metabolism pathway in FH-deficient carcinoma. Fumarate hydratase (FH) is an enzyme that converts fumarate to malate in the tricarboxylic acid (TCA) cycle in mitochondria. When FH malfunctions, fumarate and succinate accumulate, resulting in prolyl hydroxylate inhibition and consequent hypoxia-inducible factor 1-alpha (HIF-1α) stabilization. Accumulation of HIF-1α leads to the upregulation of vascular endothelial growth factor (VEGF) and glucose transporter 1 (GLUT1). VEGF promotes angiogenesis, while GLUT1 promotes glucose uptake and contributes to the etiology of high-grade FH-deficient carcinoma. In cancer cells, oxidative phosphorylation is impaired, and aerobic glycolysis (known as the Warburg effect) becomes the primary energy production mechanism. As glycolysis progresses, adenosine triphosphate (ATP) production increases, prompting cell proliferation. In FH-deficient carcinoma, the proliferation of highly malignant cancer cells depends on these mechanisms.

**Figure 2 ijms-22-07962-f002:**
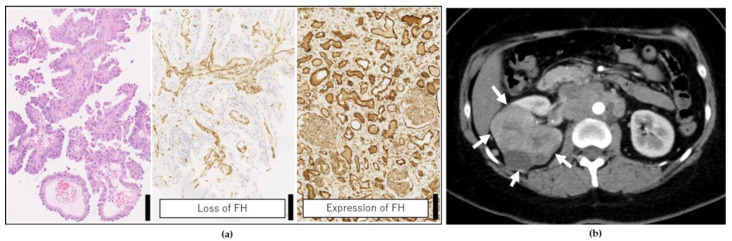
Clinical findings of the patient. (**a**) Histopathological image of the renal biopsy. The left panel shows hematoxylin-eosin staining, and the middle shows immunohistochemical (IHC) staining for FH (loss of which is evident). The right panel is the IHC staining of FH from another patient’s normal kidney as positive staining for FH. Scale bar = 100 μm. (**b**) Contrast computed tomography findings. A large tumor was found in the right kidney (white arrow), and the patient was diagnosed with stage 4 renal cell carcinoma (RCC) as well as multiple metastases in the lung, liver, supraclavicular lymph nodes, para-aortic lymph nodes, adrenal glands, and retroperitoneum.

**Figure 3 ijms-22-07962-f003:**
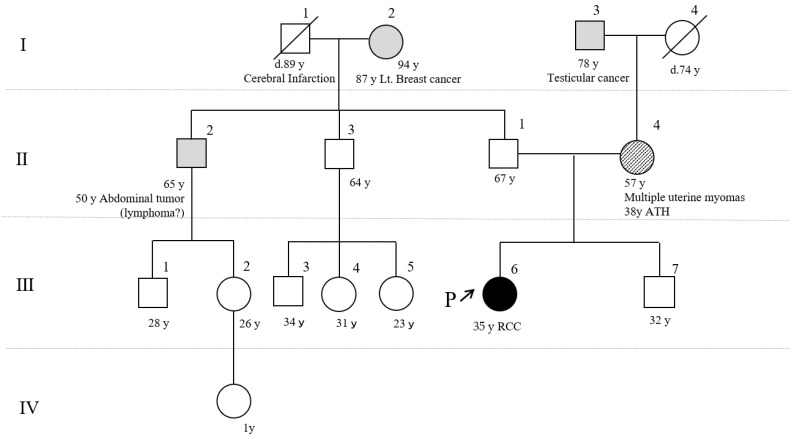
Pedigree of the patient. She (III -6) was diagnosed with RCC at the age of 35 years but had neither skin tumors nor a family history of hereditary leiomyomatosis and RCC (HLRCC). Her mother (II-4) had uterine fibroids and had undergone an abdominal total hysterectomy at age 38 years. The inherited pathological FH variant was suspected of originating from her mother. ATH, abdominal total hysterectomy; P, our patient; Roman numerals represents generation numbers; Arabic numerals represents individual numbers.

**Figure 4 ijms-22-07962-f004:**
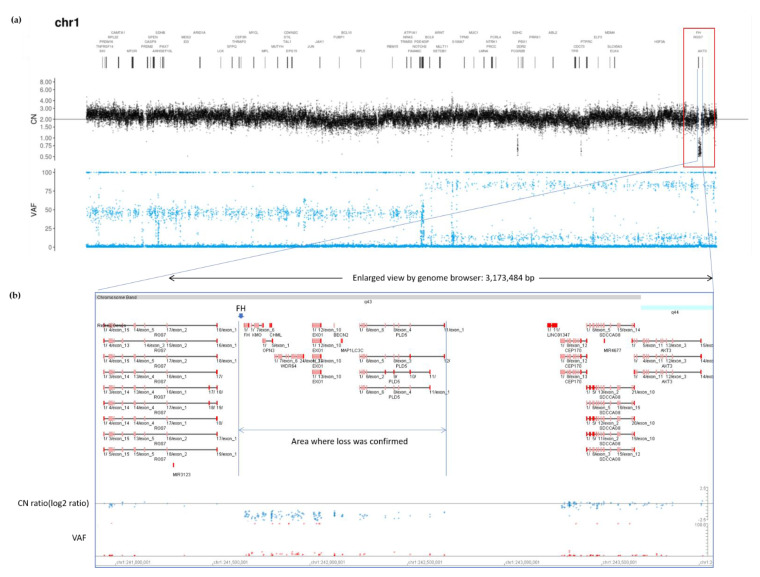
Tumor profiling data. All copy number (CN) relative ratio measurements are based on peripheral blood DNA as a control. (**a**) CN alteration (CNA) plot of full-length chromosome 1 in the tumor specimen (CN ratio of tumor tissue/normal tissue). A large homozygous deletion of approximately 3 Mb near FH was confirmed in the tumor sample. (**b**) Enlarged view using Genome Browser; a deletion was found in a large region encompassing FH. VAF, variant allele frequency; DNA, deoxyribonucleic acid.

**Figure 5 ijms-22-07962-f005:**
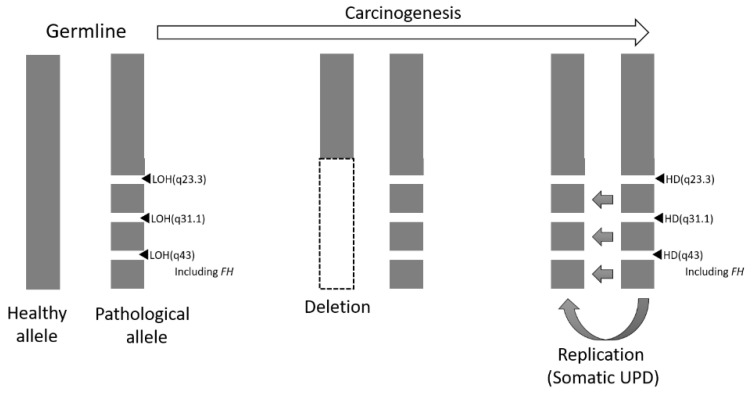
Schema of loss of FH expression during carcinogenesis. It is assumed that large-scale deletion and large-scale replication occurred during the process of transformation. LOH, loss of heterozygosity; UPD, acquired uniparental disomy; HD, homozygous deletion.

**Figure 6 ijms-22-07962-f006:**
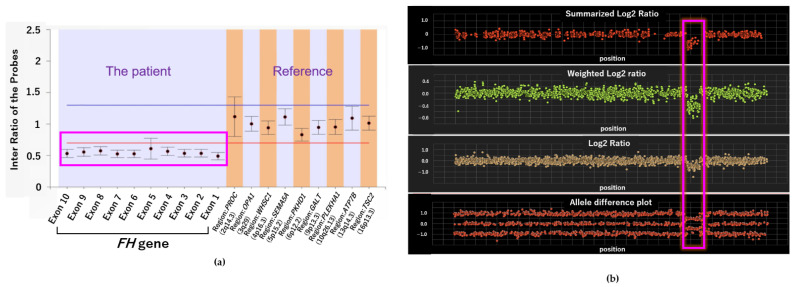
Analysis results of germline FH gene via multiplex ligation-dependent probe amplification (MLPA) and exon array. (**a**) Deletion of FH was confirmed by MLPA. (**b**) Result of exon array of chromosome 1.

**Figure 7 ijms-22-07962-f007:**
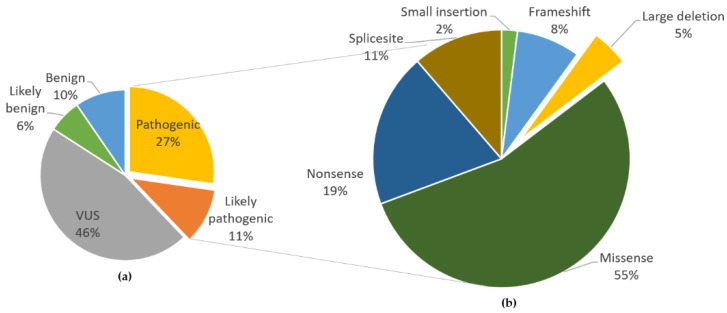
Summary of the relative frequency of FH variants. (**a**) Frequency of pathogenicity of 396 variants registered in the Leiden Open Variation Database (LOVD) [15]. Pathogenic variants comprised 27% and likely pathogenic variants 11% of all variants. (**b**) Types of pathogenic and likely pathogenic variants. Small deletions include all except whole-exon deletions, while large deletions include whole-exon to whole-gene deletions. VUS, variants of uncertain significance.

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
