# Peer review of "Germline Whole-Gene Deletion of FH Diagnosed from Tumor Profiling"

_ijms, 2021, doi:10.3390/ijms22157962_

Round 1

Reviewer 1 Report

Authors are commended for their interesting article. 

Major revisions: 

  • Authors stated that 

    The germline generic FH deletion in our patients was detected not via conventional 280 sequencing but paired tumor profiling

    They should better explain this in methods and results. I did not find sentences about the paired tumor. 

  •  

Author Response

Response to Reviewer 1 Comments

We are grateful for the feedback provided that has helped us to improve our article. We have substantially revised the manuscript according to your suggestion.

Point 1: Authors stated that: The germline genetic FH deletion in our patients was detected not via conventional sequencing but paired tumor profiling. They should better explain this in methods and results. I did not find sentences about the paired tumor. 

Response 1: I substantially modified the article. I added a detailed description of the tumor profiling.

I added a description in Introduction section;

“(line105) The European Society of Medical Oncology (ESMO) Precision Medicine Working Group recommended germline-focused analysis of tumor-only sequences in 2019, and FH was included among the 27 targeted genes [8].”

“(line111) Molecular tumor profiling is an epochal method for determining therapeutic tar-gets. Precision medicine has recently been advocated, and personalized treatment strategies based on tumor profiling have attracted attention. As a concern for tumor profiling, the possibility of hereditary cancer-related genes can be confirmed by com-parison with paired normal tissues as controls. The ACMG (2020) issued a statement on presumed germline pathogenic variants that can be revealed by tumor profiling tests [9]. They stated three tumor profiling strategies as follows: 1) tumor-only testing, 2) tumor-normal paired testing with germline variant subtraction, and 3) tumor-normal paired testing with full analysis of the germline data from a subset of genes associated with cancer predisposition [9]. We have previously developed a clinical sequencing system named “PleSSision Exome test” to perform a comprehensive tumor-normal paired profiling [10]. PleSSision Exome test analyzes both tumor and germline DNA and reports germline-derived pathogenic variants as germline findings.

By disclosing these germline findings, useful information can be provided to patients and their families, appropriate surveillance methods for early detection and early treatment for cancers can be suggested, indications for risk-reducing surgery can be discussed, and appropriate treatment for individual patients can be selected [11].”

I also added a description of PleSSision Exome test in Genetic testing result;

“(line175) As PleSSision Exome test is a tumor-normal paired profiling test, genomic deoxyribo-nucleic acid (DNA) was extracted and purified from tumor tissues as well as peripheral blood mononuclear cells (PBMCs) obtained from the patient. Exome sequencing can detect not only single nucleotide variant (SNVs) but also copy number variants (CNVs), enabling a wide range of diagnoses.”

Reviewer 2 Report

Review of the manuscript “Germline whole-gene deletion of FH diagnosed from tumor profiling” ijms-1291537

In this study Ueki at al describe a case of hereditary leiomyomatosis and renal cell carcinoma (HLRCC) caused by whole gene deletion of fumarate hydratase (HR). The patients’ germline had one good copy of chromosome 1q and one copy containing three deletions, including HR gene. The tumor genome lost the good copy of 1q which was replaced by the defective copy, resulting in complete loss of HR gene. Based on the family history the authors suspect that the defective germline copy originated from patients mother who was of UK origin. Overall this is descriptive study which is interesting and reasonably well written. However, there are some problems with the data presentation that should be addressed.

1) Figures 4, 6, S1 and S2 need to be expanded, probably to full page landscape orientation so that the genomic coordinates and genomic features are readable.

2) The Methods would deserve to be described in more detail than a single short paragraph.

3) The human genome assembly version that was used for the data analysis should be specified.

4) The authors provide no evidence that the sequencing data were deposited to any database for use by broad scientific community.

5) The case is presented as a first case described in Japan, however, since the authors suspect that the patient acquired the variant from her mother who was of UK origin, the case should be considered an import.

Author Response

Response to Reviewer 2 Comments

We are grateful for the feedback provided that has helped us to improve our article. We have substantially revised the manuscript according to each of your suggestions.

Point 1: Figures 4, 6, S1 and S2 need to be expanded, probably to full page landscape orientation so that the genomic coordinates and genomic features are readable.

Response 1: I revised the Figure 4,6, S1 and S2 according to the suggestion.

Point 2: The Methods would deserve to be described in more detail than a single short paragraph.

Response 2: I substantially modified the article. I added a detailed description of the tumor profiling and explanation of “PleSSision Exome test” in the introduction section;

“We have previously developed a clinical sequencing system named “PleSSision Exome test” to perform a comprehensive tumor-normal paired profiling [10]. PleSSision Exome test analyzes both tumor and germline DNA and reports germline-derived pathogenic variants as germline findings.”

I also added a description of PleSSision Exome test in Genetic testing result;

“As PleSSision Exome test is a tumor-normal paired profiling test, genomic deoxyribo-nucleic acid (DNA) was extracted and purified from tumor tissues as well as peripheral blood mononuclear cells (PBMCs) obtained from the patient. Exome sequencing can detect not only single nucleotide variant (SNVs) but also copy number variants (CNVs), enabling a wide range of diagnoses.”

Point 3: The human genome assembly version that was used for the data analysis should be specified.

Response 3: I added explanation on the human genome assembly version:

GRCh37 / hg19 was used in PleSSision Exome test and MLPA analysis. GRCh38 / hg38 was used in germline analysis in the OncoGuide NCC Oncopanel System FC v. 1.0.

Point 4: The authors provide no evidence that the sequencing data were deposited to any database for use by broad scientific community.

Response 4:  I added a description in Discussion;

“Our new annotated variants have been deposited to the MGeND database (https://mgend.med.kyoto-u.ac.jp/mgend.med.kyoto-u.ac.jp), which is a ClinVar-type, open-access database for use by broad scientific community.”

Sequence raw data from both PleSSision Exome test and OncoGuide NCC Oncopanel, such as BAM files, will be available upon collaborative research agreement and its approval by the research ethics committees.

Point 5: The case is presented as a first case described in Japan, however, since the authors suspect that the patient acquired the variant from her mother who was of UK origin, the case should be considered an import.

Response 5:  I added a description in Abstract, Discussion and Conclusion to clarify that this FH deletion was the first report in Japan, but it is presumed to be of maternal European origin.

Reviewer 3 Report

The case report entitled: "Germline whole-gene deletion of FH diagnosed from tumor profiling" describes the importance of recognizing each technique limitations when making a genetic diagnosis.

Major suggestions:

  1. Figure 2. - please include a panel representing normal kidney if available (it does not have to come from the matching individual, historic section could be used). It is especially important to include negative control for FH IHC, and a positive control for FH IHC staining (a normal kidney section could be used). Please include scale bars.
  2. In the methods section please include which antibody clone was used for FH IHC, as well as dilution used.
  3. Figure 6 - all of the labels are too small to read. Please enlarge all the labels so that the figure can be properly evaluated.

Minor suggestion:

Please define abbreviations at the first mention throughout the text (not only figure legends).

Author Response

Response to Reviewer 3 Comments

We are grateful for the feedback provided that has helped us to improve our article. We have substantially revised the manuscript according to each of your suggestions.

Point 1: Figure 2. - please include a panel representing normal kidney if available (it does not have to come from the matching individual, historic section could be used). It is especially important to include negative control for FH IHC, and a positive control for FH IHC staining (a normal kidney section could be used). Please include scale bars.

Response 1: I revised the Figure 2 according to the suggestion. Because the specimen was only obtained from renal biopsy, and there were few normal parts; FH expression was found in the renal tubules of the patient’s tumor. For better understanding, an FH-stained specimen of another patient's normal kidney was added to the panel. I added the scale bar in each panel.

Point 2: In the methods section please include which antibody clone was used for FH IHC, as well as dilution used.

Response 2: I added the following description of IHC staining for FH in 2.3. Materials and methods:

“Immunohistochemical analysis was performed on formalin-fixed paraffin-embedded tissues using the FH mouse monoclonal antibody (1:200 dilution; Santa Cruz Biotechnology, Santa Cruz, CA).”

Point 3: Figure 6 - all of the labels are too small to read. Please enlarge all the labels so that the figure can be properly evaluated.

Response 3: I revised the Figure 6 according to the suggestion.

Point 4: Please define abbreviations at the first mention throughout the text (not only figure legends).

Response 4:  I added the following abbreviations in the text; FH, fumarate hydratase (line 28); TCA, tricarboxylic acid (line 66); DNA, deoxyribonucleic acid (line 176); VAF, variant allele frequency (line 200).

Round 2

Reviewer 1 Report

Authors have consistently improved the manuscript. 

Author Response

We are grateful for your suggestions that have helped us to improve our article. Thank you for confirming the changes according to your  instructions. I look forward to working with you to move this manuscript closer to publication in the International Journal of Molecular Sciences.

Reviewer 2 Report

The manuscript has improved greatly. Fig 4 would still benefit from increased resolution (e.g. present it in landscape orientation).

Author Response

We are grateful for your suggestions that have helped us to improve our article. Thank you for confirming the changes according to your  instructions. I enlarged Figure4 according to your suggestion. I look forward to working with you to move this manuscript closer to publication in the International Journal of Molecular Sciences.

Reviewer 3 Report

Thank you for modifying the original manuscript according to the reviewers' suggestions.

Please include catalog number of FH antibody used for IHC.

In Figure S2, please correct "los2(tumor/blood)" to "log2" I believe.

Author Response

We are grateful for your suggestions that have helped us to improve our article. Thank you for confirming the changes according to your  instructions. I added the catalog number of FH antibody as J-13: sc-100743. And I corrected spelling mistake according to your suggestion.

I look forward to working with you to move this manuscript closer to publication in the International Journal of Molecular Sciences.